# Pathways Related to Colon Inflammation Are Associated with Colorectal Carcinoma: A Transcriptome- and Methylome-Wide Study

**DOI:** 10.3390/cancers15112921

**Published:** 2023-05-26

**Authors:** Muhammad G. Kibriya, Farzana Jasmine, Joel Pekow, Aaron Munoz, Christopher Weber, Maruf Raza, Mohammed Kamal, Habibul Ahsan, Marc Bissonnette

**Affiliations:** 1Institute for Population and Precision Health (IPPH), Biological Sciences Division, University of Chicago, Chicago, IL 60637, USA; farzana@uchicago.edu (F.J.); amunoz7@bsd.uchicago.edu (A.M.); habib@uchicago.edu (H.A.); 2Department of Public Health Sciences, Biological Sciences Division, University of Chicago, Chicago, IL 60637, USA; 3Department of Medicine, Section of Gastroenterology, Hepatology and Nutrition, University of Chicago, Chicago, IL 60637, USA; jpekow@bsd.uchicago.edu (J.P.); mbissonn@bsd.uchicago.edu (M.B.); 4Department of Pathology, University of Chicago, Chicago, IL 60637, USA; cweber@bsd.uchicago.edu; 5Department of Pathology, Jahurul Islam Medical College, Kishoregonj 2336, Bangladesh; drmarufraza@gmail.com; 6Department of Pathology, The Laboratory Dhaka, Dhaka 1205, Bangladesh; kamal.bsmmu@gmail.com

**Keywords:** colorectal carcinoma, inflammation, ulcerative colitis, nitrogen metabolism, sulfur metabolism, proteasome, IL-17, transcriptomic, methylome, RNA-seq

## Abstract

**Simple Summary:**

Colon cancer is one of the common cancers affecting a large number of patients globally. In sporadic CRC, several genetic factors may play important roles such as genetic mutations, genetic instability, and DNA promoter methylation. However, the role of inflammatory pathogenesis of sporadic CRC is less widely appreciated. Ulcerative colitis (UC) provides a unique opportunity to identify genes dysregulated in colon inflammation in human subjects and to exploit those pathways as a proxy of inflammation. In this study, we identified a number of such dysregulated metabolic and other pathways in UC and demonstrated that many of those inflammation-related gene pathways were also associated with sporadic CRC. The findings from US patients were also replicated in Bangladeshi patients with CRC. We demonstrate that this “inflammation—cancer” association was independent of sex and conventional genetic factors. The study widens our understanding of CRC pathogenesis and some of these dysregulated pathways may provide useful therapeutic options in the future.

**Abstract:**

The association of chronic inflammation with colorectal carcinoma (CRC) development is well known in ulcerative colitis (UC). However, the role of inflammatory changes in sporadic CRC pathogenesis is less widely appreciated. In this study, in the first step using RNA-seq, we identified gene-pathway-level changes in UC-associated CRC (UC CRC, *n* = 10) and used the changes as a proxy for inflammation in human colon to ask if there were associations of inflammatory pathway dysregulations in sporadic CRC pathogenesis (*n* = 8). We found down-regulations of several inflammation-related metabolic pathways (nitrogen metabolism, sulfur metabolism) and other pathways (bile secretion, fatty acid degradation) in sporadic CRC. Non-inflammation-related changes included up-regulation of the proteasome pathway. In the next step, from a larger number of paired samples from sporadic CRC patients (*n* = 71) from a geographically and ethnically different population and using a different platform (microarray), we asked if the inflammation-CRC association could be replicated. The associations were significant even after stratification by sex, tumor stage, grade, MSI status, and *KRAS* mutation status. Our findings have important implications to widen our understanding of inflammatory pathogenesis of sporadic CRC. Furthermore, targeting of several of these dysregulated pathways could provide the basis for improved therapies for CRC.

## 1. Introduction

Inflammatory bowel disease (IBD) is classified into two groups: ulcerative colitis (UC) and Crohn’s disease (CD). The annual incidence of UC in North America is 19.2 per 100,000 person-years and for CD is 20.2 per 100,000 person-years [1]. There are also indeterminate cases (IC, 10–15%), that are difficult to distinguish as UC or CD either clinically or by colon biopsy [1].

Colorectal carcinoma (CRC) is one of the most common cancers worldwide with an estimated incidence of 6.1% of all cancers [2]. The CRCs can be “hereditary”, “sporadic”, and “familial”. These terms are defined based on a family history of cancer. The term sporadic is used when there is no family history of CRC in a first-degree relative. In familial type CRC, at least one first-degree relative has CRC. Hereditary CRCs are associated with a specific inherited gene abnormality [3].

A review paper included a number of published cohort studies on CRC risk in IBD [4]. Cumulative risks of CRC were found to be 1%, 2%, and 5% after 10, 20, and >20 years of IBD, respectively [4]. A study using the Kaiser Permanente of Northern California’s database of members with IBD and general membership data showed that the incidence rates of cancer among individuals with CD, UC, or in the general membership were 75.0, 76.0, and 47.1, respectively, per 100,000 person-years. In other words, the incidence of CRC among individuals with CD or UC was 60% higher than in the general population [5].

Clinically, UC-related CRC (UC CRC) patients are younger and more frequently have multiple neoplastic lesions. Histologically, sometimes these lesions are mucinous or signet ring cell carcinomas [6]. Increased risk of CRC in IBD patients has been attributed to long-standing chronic inflammation, genetic alterations, and gut microbiome [7]. Data indicate that IBD-associated CRC may develop through a pathway of tumorigenesis distinct from that of sporadic CRC [7]. UC CRC is considered to develop progressing from a non-neoplastic inflammatory epithelium to dysplasia to cancer [6]. In a previous study, we compared UC CRC and sporadic CRC for DNA methylation and gene expression profiles [8].

Molecular mechanisms of tumorigenesis may be different in sporadic CRC and UC CRC. In sporadic CRC, several genetic factors play important roles such as genetic mutations, genetic instability, chromosomal instability (CIN) or microsatellite instability (MSI), and DNA hypermethylation. In microsatellite stable (MSS) cases, CIN results in abnormal segregation of chromosomes and abnormal DNA content (aneuploidy) contributing to the loss of function of key tumor suppressor genes such as adenomatous polyposis coli (*APC*) and *p53*. These genes can also be rendered nonfunctional by mutation [9]. Less frequently, sporadic CRCs arise through the MSI pathway. The MSI pathway involves the primary loss of function of DNA repair genes. Two of these genes, human MutL homolog-1 (*hMLH1*) and human MutS homolog-2 (*hMSH2*), are most commonly affected. These errors preferentially affect some genes that contain short nucleotide repeats in the coding regions such as transforming growth factor (*TGFRII*), *IGF2R*, and *BAX* [9]. In a previous study, we demonstrated a difference in DNA methylation pattern in MSS and MSI CRC [10].

The development of UC CRC is linked to long-standing inflammation and, unlike normal colonic mucosa, diseased colonic mucosa shows activation of inflammatory molecular pathways even before any histological evidence of dysplasia or cancer [9]. Clinical studies and animal experiments have demonstrated a crucial role of pro-inflammatory pathways, especially the NF-κB, IL-6/STAT3, COX-2/PGE2, and IL-23/Th17 signaling pathways, in the pathogenesis of UC CRC [11]. Accumulating evidence suggests that UC CRC may emerge through a distinct pathway of tumorigenesis compared to sporadic CRC [12]. UC CRC follows a sequence of genetic alterations involving “inflammation–dysplasia–carcinoma” sequence, in contrast to the “adenoma-to-carcinoma sequence” classically described in the pathogenesis of sporadic CRC. Inflammation may contribute to the development of cancer by generating reactive oxygen and nitrogen species that can cause oxidative damage to DNA [9,13]. Reactive oxygen and nitrogen species produced by inflammatory cells can interact with the protein products of cancer-related genes such as *p53*, DNA mismatch repair genes, and DNA base excision-repair genes. In addition to the strong association of inflammation in UC mucosa and UC CRC, a recent study has summarized the impact of pathogenic bacteria on colorectal inflammatory pathways [14]. For instance, Fusobacterium up-regulates microRNA-21, which is associated with the activation of IL-10 and subsequent suppression of anti-tumor T-cell-mediated immunity [15]. Fusobacterium has also been shown to trigger tumor-associated macrophages through IL-6/STAT3/c- MYC signaling [16,17].

Muller M et al. reviewed the molecular changes in UC CRC and sporadic CRC [18]. They found the mutation rate of *p53*, *KRAS*, *IDH15*, *APC* genes were significantly higher in UC CRC than the sporadic CRC. Unlike other gastrointestinal cancers, UC CRCs do not preferentially arise via a methylator pathway when compared to sporadic CRC. CpG island methylator phenotype (CIMP) was observed in 22% of sporadic CRC and in only 5% of UC CRC [19]. Hypermethylation of the *p16INK4a* promoter region is a common phenomenon that occurs early during the process of neoplastic progression in UC. Colonic hypermethylation of the *p16INK4a* occurs in 12.7%, 70%, and 100% of UC patients without dysplasia, with dysplasia, and with UC CRC, respectively [20]. Methylation of the cadherin 1 (*CDH1*) promoter was detected in 93% of IBD patients with colorectal dysplasia, versus 6% in IBD patients without dysplastic lesions. In sporadic CRC, *CDH1* promoter hypermethylation occurs in 87% of patients [21]. In another study, UC CRC tissue showed hypermethylation of several genes including *p16 INK4a*, runt-related transcription factor 3 (*RUNX3*), methylated in tumor gene 1 (*MINT1*), *MINT31*, and hyperplastic polyposis protein 1 (*HPP1*) [22].

There is strong evidence of association of inflammation and cancer in UC CRC [14,23,24]. However, this is not that well studied in sporadic CRC, which constitutes the overwhelming majority of CRC patients. UC provides a unique opportunity to identify the gene pathways dysregulated in colonic inflammation in human subjects and to exploit those gene pathways as a proxy of inflammation. In the present study, we initially investigated whether such gene pathways are dysregulated in sporadic CRC as well. In that case, inflammatory pathways could serve as a surrogate marker or indirect evidence of underlying inflammatory processes in the pathogenesis of CRC. We then examined an independent cohort of patients with sporadic CRC from different geographical backgrounds to see if our initial findings could be replicated.

## 2. Materials and Methods

US patients: Subjects were enrolled at the time of surgery and consented under University of Chicago Institutional Review Board numbers 10-209A, 15573A, and 12758A and have been described in our earlier study [8]. Subjects were included if they had either sporadic CRC or UC-associated CRC. A diagnosis of UC was confirmed by the treating gastroenterologist and histologic review by a gastrointestinal pathologist. All UC CRC used in this study were confirmed histologically to arise in an area of colon involved by UC.

Bangladeshi patients: For the replication step of this study, we used the fresh frozen tissue samples collected from Bangladeshi sporadic CRC patients from the department of Pathology, Bangabandhu Sheikh Mujib Medical University (BSMMU), Dhaka, Bangladesh, at different times spanning from December 2009 to May 2016. The patients were at different stages of CRC. From each patient, the specimens were collected from the surgically resected tumor and the surrounding unaffected part of the colon about 5–10 cm away from the tumor mass. A surgical pathology fellow collected all samples from the operating room immediately after the surgical resection. Histopathology examination was performed on H&E-stained slides from routinely processed paraffin-impregnated tissue blocks. The slides were examined independently by two pathologists and there was concordance in all cases. For staging and grading of the CRC, the World Health Organization Classification of Tumors was followed [25]. In this paper, for statistical analysis purposes, the well-differentiated tumors were categorized as “low grade” and moderately–poorly differentiated tumors were categorized as “high-grade” tumors. From each individual, we obtained a pair of tumor and normal tissues, which were frozen immediately for DNA, preserved in RNA Later (Life Technologies Corporation, Carlsbad, CA, USA), and shipped on dry ice to the molecular genomics lab at The University of Chicago for subsequent DNA, RNA extraction, and molecular assay.

For each patient, we also abstracted key demographic and clinical data and tumor characteristics from hospital medical records. Written, informed consent was obtained from all participants. The research protocol was approved by the Ethical Review Committee, Bangabandhu Sheikh Mujib Medical University, Dhaka, Bangladesh (BSMMU/2010/10096), and by the “Biological Sciences Division, University of Chicago Hospital Institutional Review Board”, Chicago, IL, USA (10-264-E).

**DNA and RNA extraction and quality control:** For the US samples, tissue was homogenized using Bullet Blender (Next Advance, Averil Park, NY, USA) and extraction was performed using the AllPrep DNA/RNA kit (Qiagen, Hidden, Germany). For the Bangladesh samples, DNA was extracted from fresh frozen tissue using Puregene Core kit (Qiagen, MA, USA). Electropherogram from Agilent BioAnalyzer with Agilent DNA 12,000 chips showed the fragment size to be >10,000 bp. RNA was extracted from RNA Later preserved colonic tissue using Ribopure tissue kit (Ambion, Austin, TX, USA, Cat# AM1924). Gene expression data were available for the first 71 patients and DNA methylation data were available for the first 125 patients.

**RNA sequencing:** RNA was prepared for sequencing using Ribo-zero rRNA Removal kit (Illumina, San Diego, CA, USA) with 1 μg of total RNA. The rRNA depleted samples were then converted to cDNA with reverse transcriptase and random priming using TruSeq RNA library preparation kit (Illumina, San Diego, CA, USA) and sequenced on HiSeq platform. The reads were aligned to human genome (Gencode GRCh37.p13 v19) using STAR aligner v2.4.0k. Details are explained in earlier study [8].

**Genome-wide gene expression assay:** We used microarray data (Illumina HT12 v4 BeadChip) from the first 71 paired tumor and normal tissue RNA (of the same set of 125 patients used for methylation assay in this study). The chip contains a total of 47,231 probes covering 31,335 genes. Paired samples were processed in the same chip (12 samples/ chip). One sample from normal tissue failed on the microarray. So, we had gene expression data from 71 CRC tissue and 70 corresponding normal tissue. Gene expression data were normalized using quantile normalization in GenomeStudio software V2011.1.

**Genome-wide methylation assay:** We used Illumina HumanMethylation450 DNA analysis BeadChip v1.0 Assay from the first 125 paired tumor-normal samples (125 pair out of the same set of 165 patients used for this study) [10]. The DNA samples were subjected to bisulfite conversion using EZ-96 DNA Methylation Kit (Zymo Research, Irvine, CA, USA). The chip presents 485,577 loci of which 150,254 were in CpG Island, 112,067 were in Shore (0–2 kb from island), 47,114 were in Shelf (2–4 kb from the island), and 176,112 were in open sea (>4 kb from CpG island). We did not include the markers in the open sea region in the final differential methylation analysis. Paired samples (CRC and corresponding normal) were processed on the same chip to avoid batch effect. From this assay, on average 17 loci per gene were investigated. A Tecan Evo robot was used for automated sample processing and the chips were scanned on a single iScan reader. If the intensity of Methylated loci is X and the intensity of unmethylated loci is Y, then, Methylation score (beta value) is X/X + Y. If all are unmethylated (X = 0), then methylation level is 0/0 + Y = 0. If all loci are methylated (Y = 0), then beta value is X/X + 0 = 1. If 50% of probes are hybridized at methylated loci and 50% are hybridized at unmethylated loci, then methylation score is 50/50 + 50 = 0.5.

**Microsatellite instability (MSI) detection:** A high-resolution melting (HRM) analysis method was used for detection of two mononucleotide MSI markers—BAT25 and BAT26 [26,27].

***KRAS* mutation detection:** Tumor and adjacent healthy colonic tissue from 165 paired (tumor and normal) tissues were tested for *KRAS* (rs 112445441) mutation by HRM analysis as described previously [26,27].

**Statistical Analysis:** To compare the continuous variables, we used t-test or one-way analysis of variance (ANOVA). Principal component analysis (PCA) and sample histograms were checked as a part of quality control analyses. Mixed-model multi-way ANOVA (which allows more than one ANOVA factor to be entered in each model) was used to compare the individual probe-level expression data (for gene expression) or beta value of CpG loci (for methylation data) across different groups. For statistical analysis, we used Partek Genomics Suite (version 7.0) (https://www.partek.com/partek-genomics-suite/ (accessed on 19 April 2023). In general, “tissue” (tumor/adjacent normal), MSI status (MSI/MSS), *KRAS* mutation (mutant/wild) were used as categorical variables with fixed effect. These levels represent all conditions of interest, whereas “person ID#” (as proxy of inter-person variation) was treated as a categorical variable with random effect since the person ID is only a random sample of all the levels of that factor. Method of moments estimation was used to obtain estimates of variance components for mixed models Eisenhart C. In the ANOVA model, the log_2_-transformed gene expression data (gene count/million read for RNA-seq or expression value for microarray) or beta-value for the CpG loci were used as the response variable (Y), and “Tumor” (tumor or normal), person ID#, “sex”, “MSI-status”, “tumor stage”, “tumor grade”, and “*KRAS* mutation status” were entered as ANOVA factors in different models.

For paired analysis, we used the following model:$$Y_{\mathrm{ijk}}=\;\mu +{\;\mathrm{Tumor}}_{i}+{\mathrm{Person}}_{j}+{\;\varepsilon }_{\mathrm{ijk}}$$
where Y_ijk_ represents the k-th observation on the i-th Tumor and the j-th Person. μ is the common effect for the whole experiment. ε_ijk_ represents the random error present in the k-th observation on the i-th Tumor and the j-th Person. The errors ε_ijk_ are assumed to be normally and independently distributed with mean 0 and standard deviation δ for all measurements. Person is a random effect.

For detection of interaction between Tumor and a factor of interest (e.g., MSI status), the following model was used:$$Y_{\mathrm{ijk}}=\;\mu +{\;\mathrm{Tumor}}_{i}+{\mathrm{MSI}\;\mathrm{status}\;\mathrm{of}\;\mathrm{CRC}}_{J}+\mathrm{Tumor}⨯{\mathrm{MSI}\;\mathrm{status}\;\mathrm{of}\;\mathrm{CRC}}_{\mathrm{ij}}+{\;\varepsilon }_{\mathrm{ijk}}$$
where Y_ijk_ represents the k-th observation on the i-th Tumor and j-th MSI status. μ is the common effect for the whole experiment. ε_ijk_ represents the random error present in the k-th observation on the i-th Tumor and j-th MSI status of CRC. The errors ε_ijk_ are assumed to be normally and independently distributed with mean 0 and standard deviation δ for all measurements.

Gene Ontology (GO) was used to group a set of genes into a category. In GO Enrichment analysis, we tested if the genes found to be differentially expressed or methylated fell into a Gene Ontology category more often than expected by chance [28]. We used chi-square test to compare “number of significant genes from a given category/total number of significant genes” vs. “number of genes on chip in that category/total number of genes on the microarray chip”. Negative log of the p-value for this test was used as the enrichment score. Therefore, a GO group with a high enrichment score represents a lead functional group. The enrichment scores were analyzed in a hierarchical visualization and in tabular form.

Gene set ANOVA is a mixed model ANOVA to test the expression or methylation of a set of genes (sharing the same category or functional group) instead of an individual gene in different groups (https://www.partek.com/partek-genomics-suite/ (accessed on 19 April 2023). The analysis is performed at the gene level, but the result is expressed at the level of the gene set category by averaging the member genes’ results. The equation for the model was:Model: Y = μ + T + P + G + S (T × P) + ε
where Y represents the expression or methylation status of a Gene set category, μ is the common effect or average expression/methylation of the Gene set category, T is the tissue-to-tissue (tumor/normal) effect, P is the patient-to-patient effect, G is the gene-to-gene effect (differential expression or methylation of genes within the gene set category independent of tissue types), S (T × P) is the sample-to-sample effect (this is a random effect and nested in tissue and patient), and ε represents the random error.

## 3. Results

The patient characteristics of the UC CRC patients (male = 10, female = 0) and sporadic CRC patients (male = 2, female = 6) are shown in Appendix A. All of the patients were from the University of Chicago Medical Center and were included in our prior study [8]. From these two groups of patients, we had four types of tissues (see Figure 1) from which we extracted RNA and which we used for RNA-seq experiments. Due to the nature of colonic involvement in UC (pancolitis in all cases), it was not possible to obtain a section of healthy colonic mucosa. Therefore, in this study, we considered the sections from adjacent non-cancerous tissue of sporadic CRC patients (example Figure 1A) as “Control” for comparison purposes.

### 3.1. Inflammation-Related Differentially Expressed Gene Pathways in Colon Tissue

Considering the nature of pancolitis in UC, to identify the gene pathways altered in inflammation, we compared the expression profile of adjacent non-cancer UC colonic mucosa from the UC CRC patients (*n* = 10, example Figure 1C) and adjacent non-cancer colonic mucosa (Control) of sporadic CRC patients (*n* = 8, example Figure 1A). The total list of 53 differentially expressed Kyoto Encyclopedia of Genes and Genomes (KEGG) pathways are shown in Appendix A. All were statistically significant at FDR 0.05 and had ≥2-fold change (FC). Some of the differentially expressed pathways are shown in the Figure 2 including pathways related to sulfur metabolism (Figure 2A), nitrogen metabolism (Figure 2B), peroxisomes (Figure 2C), bile secretion (Figure 2D), drug metabolism (Figure 2E), Riboflavin metabolism (Figure 2F), fatty acid degradation (Figure 2G), and mineral absorption (Figure 2H).

### 3.2. Differentially Expressed Gene Pathways in UC CRC

To identify the gene pathways altered in UC CRC patients, we compared the expression profile of cancerous UC colonic mucosa (UC CRC) from the UC CRC patients (*n* = 10, example Figure 1D) and adjacent non-cancer colonic mucosa (Control) of sporadic CRC patients (*n* = 8, example Figure 1A). The total list of 61 differentially expressed KEGG pathways are shown in Appendix A. In addition to the pathway level analyses, we also analyzed the data at gene-level and looked for enrichment of the significant genes (Appendix A).

### 3.3. Differentially Expressed Gene Pathways in Sporadic CRC

To identify the gene pathways altered in sporadic CRC patients, we compared the expression profile of cancerous colonic mucosa (Sporadic CRC) from the sporadic CRC patients (*n* = 8, example Figure 1B) and adjacent non-cancer colonic mucosa (Control) of sporadic CRC patients (*n* = 8, example Figure 1A). The total list of 59 differentially expressed KEGG pathways are shown in Appendix A.

### 3.4. Association of Colonic Mucosal Inflammation and CRC

In the next step, to figure out the involvement of differentially expressed gene pathways in inflammation and CRC (UC CRC or Sporadic CRC), we used the Venn diagram (Figure 3A) of the three lists described above.

The Venn diagram (Figure 3A) shows that out of the 53 inflammation-related pathways (pink in Figure 3A), about 81% of them (29 + 14 = 43 out of 53) were found to be differentially expressed (all down-regulated) in UC CRC compared to the control group. Among 61 differentially expressed pathways in UC CRC (light blue in Figure 3), 70% of them (29 + 14 = 43 out of 61) were already down-regulated in adjacent UC mucosa without CRC. The finding suggests that, at least in these patients, the UC disease process (inflammation) primed the colonic mucosa for the development of CRC.

When we compared the UC CRC tissue (*n* = 10, example Figure 1D) to corresponding non-cancerous UC tissue (*n* = 10, example Figure 1C), only four pathways were differentially expressed (see Figure 3B). The three up-regulated pathways were DNA replication, homologous recombination, and proteasome (see Figure 4, these were not inflammation related). The only one down-regulated pathway was the nitrogen metabolism pathway, which was inflammation-related (and was down-regulated in UC compared to control).

The overlap of differentially expressed pathways in sporadic CRC (see Figure 3A) also suggested that about 49% (29 out of 59) of the differentially expressed gene pathways in sporadic CRC are also found to be associated with inflammatory changes in the colon—further emphasizing the role of inflammation in sporadic CRC pathogenesis. The detailed results of these common 29 differentially expressed pathways in sporadic CRC and UC CRC are presented in Appendix A respectively. All of these 29 pathways were down-regulated in both sporadic CRC and UC CRC. A few of them are shown in Figure 5.

### 3.5. Replication Study in Sporadic CRC

In the next step, we tried to see if these associations of inflammation-related pathways and sporadic CRC found in US patients could be replicated in an independent set of CRC patients of a different geographical background. We utilized microarray gene expression data (Illumina HG12v4) from one of our previous studies [29,30] involving Bangladeshi patients with sporadic CRC (BD sporadic CRC). All the patients (males = 43, females = 28) were native Bangladeshi and we used paired tissue samples (tumor and adjacent normal from the same patient) for comparison. Patient characteristics are shown in Appendix A.

We used a similar Gene set ANOVA for the KEGG pathways. Out of the 29 inflammation-related gene pathways that we identified in US patients, we could not analyze only one pathway (Neuroactive ligand-receptor interaction) in this BD sporadic CRC patients. Results of the 28 pathways are shown in Appendix A. It was interesting to note that in general 25 of these 28 pathways were significantly down-regulated (at FDR < 0.05) in CRC tissue compared to the paired normal tissue. Figure 6 shows examples of PCA plots based on gene expression level of nitrogen metabolism genes (Figure 6A) and fatty acid degradation genes (Figure 6B) without selecting the individual genes by statistical significance. There was reasonable separation between CRC tissues and adjacent normal tissues.

We also carried out stratified analyses by (a) sex (Appendix A), (b) tumor stage (Table 1), (c) tumor grade (Appendix A), (d) MSI status (Table 2), and by (e) *KRAS* mutation status (Appendix A). The analyses suggested that most of these inflammation-related pathways were significantly down-regulated in CRC tissue compared to paired normal tissue irrespective of sex, stage, grade, MSI status or *KRAS* mutation status.

For some of the pathways, the magnitude of down-regulation was similar in all stages (without statistical difference, shown as interaction *p*-value in Table 1). Nitrogen metabolism is an example of this category (see Figure 7 upper panel)—the genes in this pathway were down-regulated by 72%, 76%, and 66% in stage 1, stage 2, and stage 3 CRC tissue, respectively, compared to their paired normal tissue (interaction *p* = 0.55, see Table 1). However, for some pathways, the magnitude of down-regulation was slightly but significantly more pronounced in advanced stages compared to stage 1. If we look at fatty acid degradation pathway as an example (see Figure 7 lower panel), compared to corresponding adjacent normal colonic tissue, the genes in this pathway were overall down-regulated by 9%, 21%, and 19% in stage 1, stage 2, and stage 3 CRC, respectively (interaction *p* = 1.5 × 10^−7^) (see Table 1). When the patients were stratified by grade, the same fatty acid degradation pathway was down-regulated by 17% in low grade CRC compared to 22% in high grade CRC patients (interaction *p* = 9.05 × 10^−5^, see Appendix A). When stratified by MSI status, these genes of fatty acid degradation pathway were down-regulated by 21% in presence of MSI in CRC patients compared to 17% in MSS patients (interaction *p* = 5.4 × 10^−11^, see Table 2). However, the magnitude of down-regulation was similar in *KRAS* mutant CRC (17%) and *KRAS* wild type CRC (17%) (interaction *p* = 0.33, see Appendix A).

We also asked the question if this inflammation-related dysregulation (in this case down-regulation) that we observed in sporadic CRC tissue was from the tumor infiltrating leucocytes (TIL). Although admitting the fact that we only had bulk RNA-seq or microarray data and not the single cell RNA-seq data, we still examined the data to see if the down-regulation was different in magnitude in the absence or presence of TIL. The result is presented in Appendix A. The magnitude of down-regulation in the majority of these inflammation-related pathways was not different by TIL status.

The list of differentially expressed genes that contributed to the dysregulation of the several inflammation-related pathways in sporadic CRC are presented in Appendix A.

### 3.6. Nitrogen Metabolism, Genes for Urea Cycle Enzymes

In light of the down-regulation of nitrogen metabolism in CRC tissue, we further explored the relevant genes for urea cycle. Our data suggested that, in CRC, the up-regulation of carbamoyl phosphate synthase 1 (*CPS1*) (FC = 1.56 (95% CI 1.16–2.10), see Figure 8A) might help cancer cells to convert the toxic NH4 to carbamoyl phosphate (CP) that could be utilized for pyrimidine synthesis for growth and proliferation. Down-regulation of ornithine transcarbamylase (*OTC*) (FC= −1.52 (95% CI −1.74 to −1.32), see Figure 8B) reduces the citrulline within mitochondria but increases ornithine in the cytoplasm, which helps in cancer initiation and progression [23]. While one of the isomers of argininosuccinate synthase (*ASS1*) is slightly increased (FC = 1.06 (95% CI 1.03–1.09), see Figure 8C), the down-regulation of argininosuccinate lyase (*ASL*) (FC = −1.24 (95%CI −1.33 to −1.15), see Figure 8D) may decrease the generation of fumarate, which is needed for mitochondrial metabolism and thus pushes the cancer cell from aerobic to anaerobic metabolism. Up-regulation of NO synthase (*NOS3*) (see Figure 8E) increases the arginine-citrulline cycle and thus enables the cancer cell to avoid more urea production and to rather utilize the nitrogen to meet the increased demands for nucleic acid. We did not find any differential expression of the gene arginase1 (*ARG1*) which produces the enzyme arginase (*ARG*), that helps in conversion of arginine to ornithine (FC = 1.02 (95% CI −1.009 to 1.06, see Figure 8F).

### 3.7. Differential Expression of Previously Reported Inflammatory Genes in Our Dataset

A previous study showed several chemokine-related genes to be dysregulated in adenomas [31]. We checked if those genes are differentially expressed in carcinomas as well. Our data confirm the gene expression changes in the same direction for CRC (Figure 9). Another study suggested down-regulation of *SLC4A4* and up-regulation of *NMUR1*, *TIMP1,* and *TACR3* occurs in CRC tissue [32]. Our data also showed a 4-fold down-regulation of *SLC4A4*, and a 2-fold up-regulation of *TIMP1* in CRC tissue compared to corresponding normal tissue. Other than the gene pathways identified from the UC patients, when we looked at differential expression of all the KEGG pathways, IL-17 pathway was one of the top pathways that was up-regulated in sporadic CRC, and Th1 and Th2 cell differentiation pathway was down-regulated in CRC (Figure 9). In fact, previous studies suggested *PTGS2* and *IL17* may be important drivers of inflammation in sporadic CRC [33,34].

### 3.8. DNA Methylation of These Inflammation-Related Genes in CRC

We next looked to see if the aforementioned inflammation-related gene pathway dysregulation might be driven by dysregulation in DNA promoter methylation. To this end, we used our DNA methylation data (Infinium Human Methylation 450 K BeadChip array) from our previous study [10] involving 250 samples from 125 CRC patients (males = 71, females = 54). The patient characteristics are shown in Appendix A. Of these 125 CRC patients, 71 had gene expression data that we have used for the gene pathway analysis above.

The cross-tabulation of the methylation markers in the autosomes by functional group and relation to CpG island is shown in Appendix A. In this study, we focused only on the promoter-associated markers in the CpG islands. We employed a Gene set ANOVA similar to that we used for gene expression. The promoter methylation data for the same 28 pathways are shown in Appendix A. Differential methylation is shown as delta beta (beta value of CRC tissue—beta value of corresponding adjacent normal tissue) where positive values reflect hyper-methylation and negative values reflect hypo-methylation in CRC tissue. The analysis suggested that overall hyper-methylation of genes in a few pathways (ovarian steroidogenesis, taste transduction, folate biosynthesis, bile secretion, and pentose and glucuronate interconversions; see Figure 10) might explain the observed down-regulated genes in those pathways seen in the gene expression analysis above.

## 4. Discussion

In this study, we investigated the association of differential gene expression changes at pathway level in inflammatory conditions and in colon cancer. From the RNA sequencing data of a small number of patients, we identified a number of gene pathways that are dysregulated (all were down-regulated) in UC and were also found to be differentially expressed in UC CRC as well as sporadic CRC. In the next step, from a larger number of sporadic CRC patients from a geographically different population, using a different platform (microarray), we examined if those pathways dysregulated in chronic inflammatory conditions (UC) were also dysregulated in CRC tissue compared to paired adjacent normal colon tissue. The analyses validated our initial finding. We also show that this association of inflammation and CRC was seen irrespective of sex, CRC stage, grade, MSI status or *KRAS* mutation status of the tumor.

In our study, we showed that several colonic inflammation-related gene pathways were associated with sporadic CRC pathogenesis. These genes are related to metabolic pathways including sulfur metabolism (*PAPSS2*, *ETHE1*, *TST,* and *MST)*, nitrogen metabolism (*CA2*, *CA4,* and *CA12*), ascorbate metabolism, drug metabolism, tyrosine metabolism, tryptophan metabolism, and bile-secretion-related (*SLC4A4*, *AQP8*) and chemical-carcinogen-related (*UGT1A10*, *NAT2*) pathways. We found that all of these pathways were down-regulated in CRC.

Sulfur metabolism plays an important role in sulphomucin production. Sulphomucin protects intestinal mucosa. A previous study suggested that 3′-phosphoadenosine-5′-phosphosulfate (PAPS), a sulfate donor, is generated from adenosine trisphosphate (ATP) and inorganic sulfate (SO_4_^2−^), in which the PAPS synthase 2 (PAPSS2) is the key catalytic enzyme [35]. A recent study showed that expression of *PAPSS2* gene is decreased in the colon cancers of mice and humans, and deletion of PAPSS2-PAPS made mouse models more susceptible to malignant transformation [35]. Their results are consistent with our findings. The PAPSS2-PAPS-sulfation axis plays an essential role in colitis and colonic carcinogenesis. In one study, the authors propose that intestinal sulfation may represent a potential diagnostic marker, and PAPSS2 may serve as a potential therapeutic target for IBD and colon cancer [36]. Hydrogen sulfide (H_2_S) is produced by other metabolic processes involving the microbiome and inhibits oxidative phosphorylation. ETHE1 is the enzyme responsible for H_2_S breakdown leading to increased aerobic glycolysis (the Warburg effect), oxidative phosphorylation, and mitochondrial biogenesis. Investigators found higher levels of ETHE1 enzyme in normal colon epithelium adjacent to tumors [37]. Both in UC CRC and sporadic CRC tissue, we found the decreased expression of *ETHE1* gene. *TST* and *MST* expression are both increased during colonocyte differentiation. *TST* is responsible for H_2_S catabolism. Reduced *TST* expression and activity would be predicted to increase H_2_S, which could cause cell loss and inflammation that occurs in UC and CRC [38]. Levels of both enzymes (product of *TST* and *MST*) was focally lost in UC and markedly reduced in advanced colon cancer [38], which supports our finding of inflammation-associated pathways in CRC.

Genes associated with nitrogen metabolism have been studied by other investigators and their findings are consistent with our results [39,40,41,42]. A study showed that Carbonic anhydrase 2 (*CA2*) was down-regulated in UC mucosa compared to normal colonic mucosa. *CA2* expression was almost undetectable in UC CRC [39]. The carbonic anhydrases CA1, CA2, CA4, and CA12 were found to be less abundant in UC [40]. In other cancers, such as cervical cancer, immunohistochemical expression of CA12 protein was strongly associated with the histologic grade of cervical cancer. A lack of CA12 protein expression was associated with the poorly differentiated type [42]. CRC patients with higher expression of CA2 have a better prognosis [41]. Our current study confirms that, in human CRC, there is down-regulation of genes involved in nitrogen metabolism genes in CRC. A recent review paper has discussed the potential of targeting nitrogen metabolism for cancer therapy [43]. Molecular data from our study in human subjects with CRC also suggest a rationale for targeting nitrogen metabolism in CRC therapy.

We also found genes related to the bile acid secretion pathway to be down-regulated in CRC. Several genes in this pathway have been reported previously. For example, recent studies using immunohistochemistry showed that AQP8 expression was low in adenocarcinoma of the colon and higher levels of AQP8 were significantly associated with better survival in CRC patients [44,45]. Our study also found similar results. Another study using transcriptomes of two cohorts, GSE41258 and GSE32323, contained in The Cancer Genome Atlas (TCGA) analyzed differences in *SLC4A4* expression between tumor and normal tissue, showed that *SLC4A4* expression was lower in colon adenocarcinoma than in normal colon tissue, and suggested lower expression was associated with poor prognosis. Reduced *SLC4A4* expression was also associated with lymph node invasion and distant metastasis [46].

We also identified down-regulation of the chemical carcinogenesis pathway in CRC. In this respect, the inter-individual variation in *NAT1* and *NAT2* in the colon could affect how individuals respond to exposure to specific NAT substrates including carcinogens [47]. Injury to the colorectal mucosa caused by environmental carcinogens or any other agents damaging the mucosa could elicit an inflammatory process that could increase the risk of malignant transformation [48]. At least one study, however, did not find an association between *NAT1* and *NAT2* polymorphisms and IBD or sporadic CRC [49]. Wang et al. showed significantly low positivity of UGT1A in adenocarcinoma using immunohistochemistry in normal colonic mucosa, adenoma, and adenocarcinoma [50].

The main objective of this study was to look for associations between inflammation and CRC. UC represents an ideal disease state to identify gene pathways dysregulated in chronic inflammation of the colon. With that in mind, we identified a number of pathways that were grossly dysregulated (FDR ≤ 0.05 and FC ≥ 2) in UC mucosa compared to the control (adjacent not affected colonic mucosa from CRC patient). We used dysregulation of those pathways as proxies for chronic inflammation in colon and tested the association with CRC. From this perspective, one of the drawbacks of our study is that we only included UC patients who had developed CRC. This cohort may represent more severe cases of UC. Thus, we did not look for early inflammatory changes or dysregulation in UC. However, comparing the non-cancerous UC mucosa from these patients to the CRC tissue from the same individual provides a better opportunity to identify the pathways that lead to the development of CRC in UC patients by eliminating inter-individual differences. In fact, our data suggest that the inflammatory process in the UC would predispose CRC development. The up-regulated homologous recombination pathway in UC CRC tissue compared to adjacent UC tissue (Figure 4C) indicated activation of DNA damage response in UC CRC, and there was significant up-regulation of DNA replication pathway (Figure 4B), which is consistent with carcinogenesis in this situation. The up-regulation of genes in the proteasome pathway in the UC CRC tissue (Figure 4A) compared to non-cancerous UC tissue may suggest a potential role of proteasome inhibitor (bortezomib, carfilzomib, or ixazomib) in UC CRC patients if necessary. In fact, the proteasome pathway was also one of the most significantly up-regulated pathways in BD sporadic CRC patients, suggesting a potential role of proteasome inhibitor in sporadic CRC. These medications are used in multiple myeloma but, to our knowledge, have not yet been tried in UC CRC or sporadic CRC. Potential toxicity is concerned; however, based on the molecular data, in refractory cases, it may be considered for future study.

Another limitation of the study may be the small number of UC samples; however, the inflammation-related pathways identified from these samples were found to be associated with sporadic CRC not only in US patients but were also replicated in a larger number of patients from a different country and using a different platform. In fact, this may be considered as a strength of the finding. The down-regulated gene pathways (proxies of inflammatory process) that we found in sporadic CRC are very unlikely to be a direct effect of any bacterial or fungal infection, as these were found in CRC tissue compared to paired normal colonic mucosa. With all the limitations of our study in mind, some of the strengths of the study may be noted. First, paired tumor–normal samples from the same individual for comparison is a robust method for detecting any gene expression and methylation changes in cancer. Second, we used tissue samples preserved in RNA Later for RNA and fresh frozen sample for DNA, which are the gold standards for such assays. Third, to our knowledge, this is one of the first studies to comprehensively address the association of inflammation and pathogenesis of human sporadic CRC with replication in a different population.

Our study identified a number of gene pathways that were down-regulated in association with colonic inflammation. Their down-regulations may contribute to inflammation and CRC development not only in UC patients, but also in sporadic CRC. In sporadic CRC cases, during colonoscopy, we generally do not see gross inflammation in unaffected parts of the colon. However, these pathways were also down-regulated in the tumor tissue (CRC), perhaps contributing to a molecular inflammatory process, as detected by gene expression. The role of inflammation in carcinogenesis in UC CRC is widely known, but our study suggests an association of inflammation with sporadic CRC pathogenesis as well, irrespective of some known drivers such as *KRAS* mutation and MSI.

## 5. Conclusions

Our study demonstrates an association between chronic inflammatory processes in the colonic mucosa and the development of sporadic CRC. This widens our understanding of the inflammatory pathogenesis of sporadic CRC, and some of these dysregulated pathways may provide useful therapeutic options in the future.

## Figures and Tables

**Figure 1 cancers-15-02921-f001:**
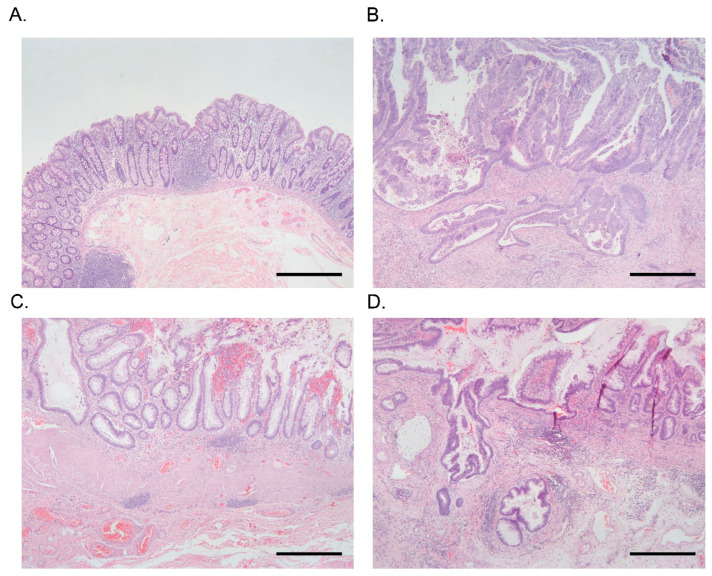
Photomicrographs of H&E stain of colonic tissues from a patient of sporadic CRC (upper panel (**A**,**B**)) and from a patient of UC CRC (lower panel (**C**,**D**)). Scale bar 500 µm in all four images: (**A**) Normal colonic mucosa and submucosa. No dysplasia is present. (**B**) Invasive, moderately differentiated adenocarcinoma arising in a patient with sporadic CRC without inflammatory bowel disease. Tumor glands are invading into the submucosa. (Same patient as (**A**)). (**C**) Colonic mucosa and submucosa in a patient with active ulcerative colitis. No dysplasia or carcinoma is present. The mucosa demonstrates features of architectural distortion and presence of moderate active (neutrophilic) inflammation, and there is some thickening of the muscularis mucosa with mild superficial submucosal fibrosis. (**D**) Invasive, moderately differentiated adenocarcinoma arising in a patient with ulcerative colitis. There is in situ dysplasia in the mucosa overlying the tumor glands present within the submucosa. (Same patient as (**C**)).

**Figure 2 cancers-15-02921-f002:**
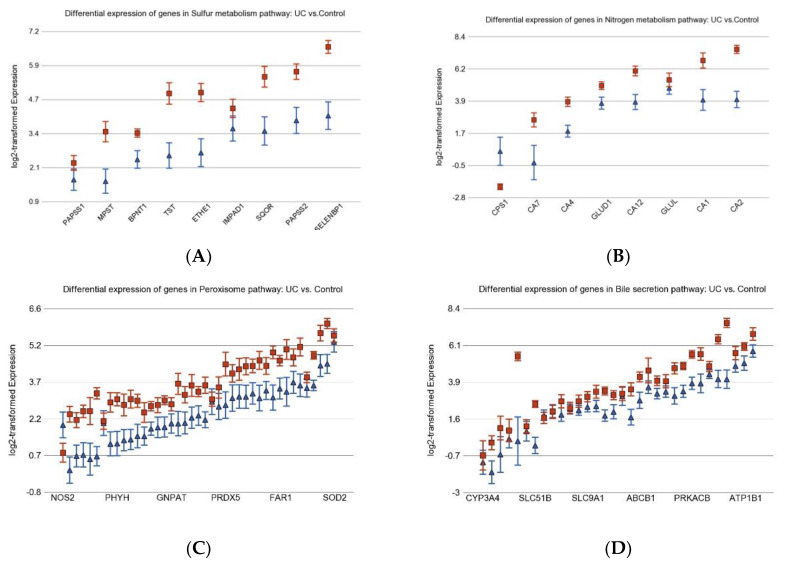
Differentially expressed gene pathways in UC (in blue triangles) compared to Control (in red squares). The genes are arranged on the *x*-axis by expression level and the log_2_-transformed expression in counts-per-million-aligned reads is shown on the *y*-axis. Gene symbols for all the genes in a given pathway could not be presented in the figure because of space limitation. A difference of 1 on the *y*-axis represents 2-fold change.

**Figure 3 cancers-15-02921-f003:**
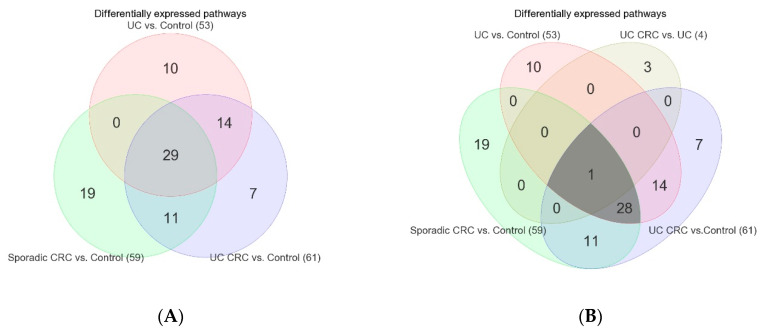
Venn diagram: (**A**) intersection of lists of differentially expressed pathways in (a) sporadic CRC vs. Control (green), (b) UC CRC vs. Control (blue), and (c) UC vs. Control (pink); (**B**) intersection of lists of differentially expressed pathways in (a) sporadic CRC vs. Control (green), (b) UC CRC vs. Control (blue), (c) UC vs. Control (pink), and (d) UC CRC vs. UC (light yellow).

**Figure 4 cancers-15-02921-f004:**
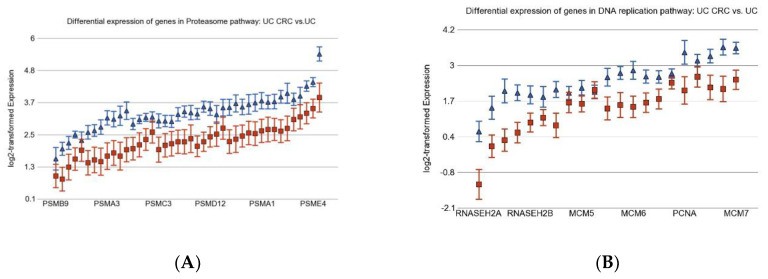
Gene expression pathways differentially expressed in UC CRC (in blue triangles) compared to adjacent non-cancerous UC mucosa (in red squares). The genes are arranged on the *x*-axis by expression level; gene symbols for all the genes in the pathway could not be presented in the figure because of space limitation. Log_2_-transformed expression is shown on *y*-axis. A difference of 1 on the *y*-axis represents 2-fold change.

**Figure 5 cancers-15-02921-f005:**
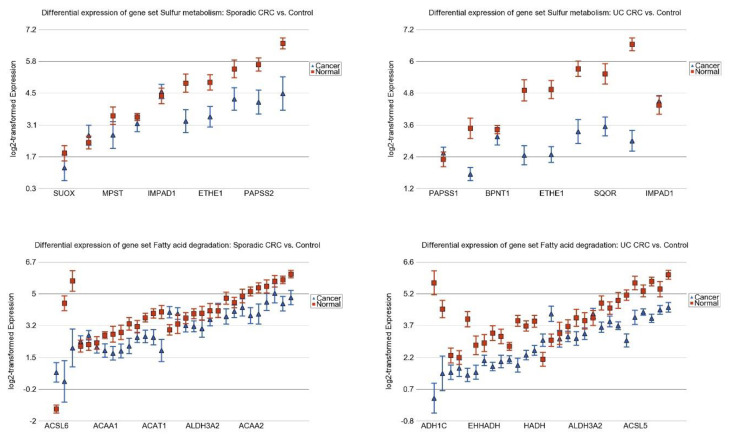
Colonic mucosal inflammation-related pathways that are down-regulated in both sporadic CRC (**left panel**) and UC CRC (**right panel**). Cancer tissues (for both sporadic CRC and UC CRC) are shown in blue triangles and the normal tissues/control are shown in red squares. The genes within the pathways are arranged on the *x*-axis by expression level; gene symbols for all the genes in the pathway could not be presented in the figure because of space limitation. Log_2_-transformed expression is shown on the *y*-axis. A difference of 1 on *y*-axis represents 2-fold change.

**Figure 6 cancers-15-02921-f006:**
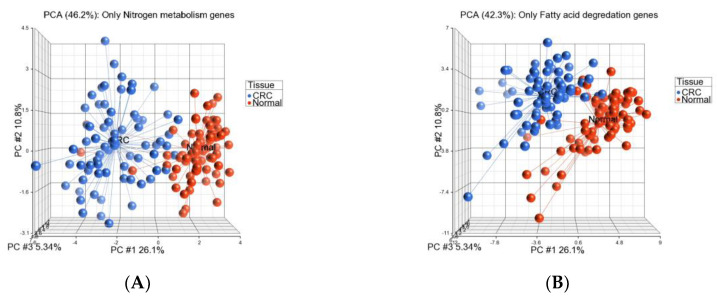
PCA plot based on nitrogen metabolism genes (shown in (**A**)) and fatty acid degradation genes (shown in (**B**)) in Bangladeshi sporadic CRC patients. CRC tissue is shown in blue and adjacent normal tissue is shown in red.

**Figure 7 cancers-15-02921-f007:**
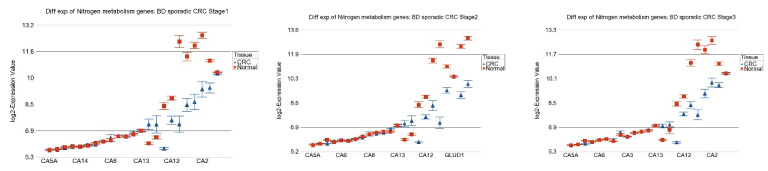
Differential expression of nitrogen metabolism genes (**upper panel**) and fatty acid degradation genes (**lower panel**) stratified by tumor staging in the replication cohort of BD sporadic CRC patients. Stage 1 is shown on the left, stage 2 in the middle, and stage 3 on the right panel. Sporadic CRC tissue is shown in blue and adjacent normal tissue is shown in red.

**Figure 8 cancers-15-02921-f008:**
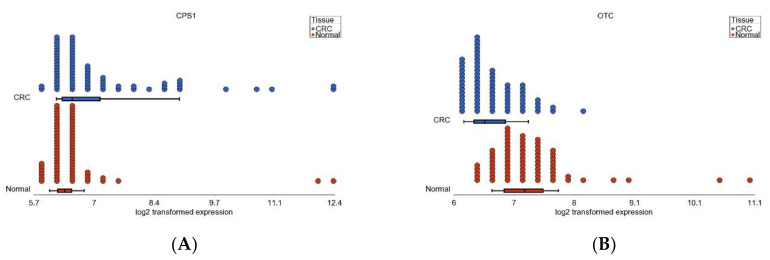
Differential expression of genes coding for urea cycle enzymes. The dot plots show the distribution of the samples (CRC tissue in blue, while normal tissue in red). Log_2_-transformed expression of the gene probes are shown on the horizontal axis. *CPS1* (shown in **A**) and *OTC* (shown in **B**) code for mitochondrial enzymes. The other genes code for cytosolic enzymes. *ASS1* (shown in **C**) converts citrulline to argininosuccinic acid (ASA). *ASL* (shown in **D**) helps synthesize fumarate from ASA. Nitric acid synthase3 (*NOS3*) is needed to convert arginine to citrulline (shown in **E**). The gene arginase1 (*ARG1*) (shown in **F**) produces the enzyme arginase (*ARG*), which helps in conversion of arginine to ornithine.

**Figure 9 cancers-15-02921-f009:**
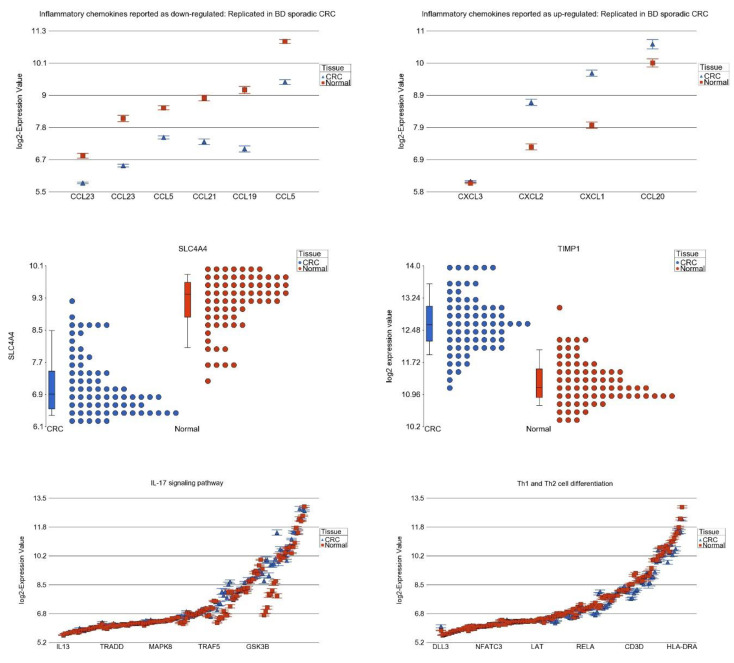
Differential expression of previously reported down-regulated chemokine-related genes (**upper panel, left side**) and up-regulated chemokine-related genes (**upper panel, right side**), down-regulation of SLC4A4 (**middle panel, left side**) and up-regulation of TIMP1 (**middle panel, right side**). Up-regulation of expression of IL-17 signaling pathway (**lower panel, left side**) and down-regulation of Th1 and Th2 cell differentiation gene (**lower panel, right side**). Sporadic CRC tissue is shown in blue and adjacent normal tissue is shown in red.

**Figure 10 cancers-15-02921-f010:**
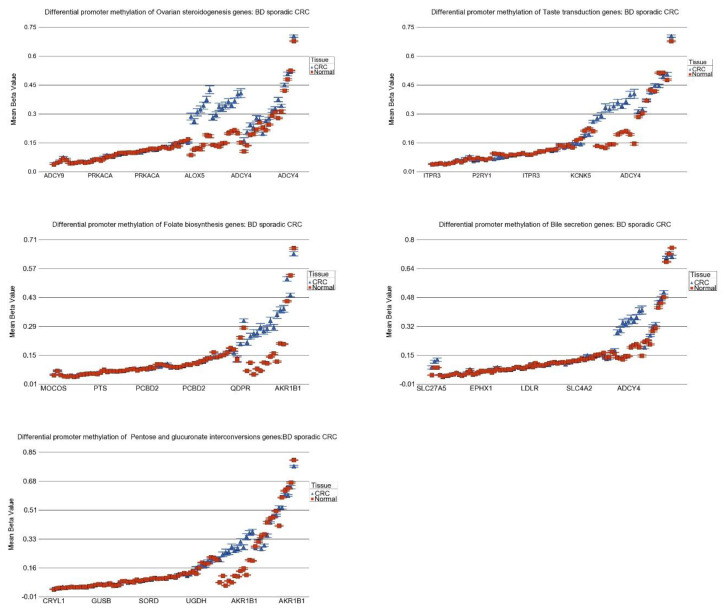
Differential DNA promoter methylation of inflammation-related gene pathways. The genes within the pathways are arranged on the *x*-axis by methylation level; there are multiple probes for a gene, and gene symbols for all the genes in a given pathway could not be presented in the figure. Methylation Beta value is shown on the *y*-axis. CRC tissue is shown in blue, and the corresponding adjacent normal tissue is shown in red.

**Table 1 cancers-15-02921-t001:** Differential expression of the 28 colon-inflammation-related pathways (identified from US sporadic CRC patients), replicated in BD sporadic CRC patients stratified by CRC stage.

Pathway (GO Description)	Stage 1 vs. Normal		Stage 2 vs. Normal		Stage 3 vs. Normal		Interaction *p*
	FC	(95% CI)	FC	(95% CI)	FC	(95% CI)	
Nitrogen metabolism	−1.72	(−1.94–−1.53)	−1.76	(−1.92–−1.61)	−1.66	(−1.79–−1.55)	0.558
Proximal tubule bicarbonate reclamation	−1.27	(−1.34–−1.20)	−1.38	(−1.44–−1.32)	−1.34	(−1.38–−1.29)	1.29 × 10^−3^
Pentose and glucuronate interconversions	−1.21	(−1.28–−1.15)	−1.32	(−1.37–−1.27)	−1.24	(−1.28–−1.20)	0.084
Retinol metabolism	−1.20	(−1.24–−1.16)	−1.27	(−1.30–−1.23)	−1.19	(−1.22–−1.17)	8.33 × 10^−3^
Sulfur metabolism	−1.20	(−1.29–−1.12)	−1.31	(−1.39–−1.24)	−1.30	(−1.36–−1.24)	3.65 × 10^−2^
Synthesis and degradation of ketone bodies	−1.18	(−1.28–−1.09)	−1.26	(−1.34–−1.19)	−1.22	(−1.28–−1.16)	0.353
Mineral absorption	−1.17	(−1.21–−1.14)	−1.16	(−1.19–−1.14)	−1.17	(−1.19–−1.15)	1.10 × 10^−7^
Drug metabolism-cytochrome P450	−1.14	(−1.18–−1.10)	−1.20	(−1.23–−1.17)	−1.20	(−1.22–−1.17)	9.76 × 10^−3^
Ascorbate and aldarate metabolism	−1.13	(−1.19–−1.06)	−1.30	(−1.35–−1.24)	−1.21	(−1.25–−1.17)	9.02 × 10^−4^
Steroid hormone biosynthesis	−1.13	(−1.17–−1.09)	−1.19	(−1.22–−1.16)	−1.16	(−1.18–−1.13)	0.119
Chemical carcinogenesis	−1.11	(−1.14–−1.08)	−1.17	(−1.19–−1.15)	−1.17	(−1.19–−1.15)	2.70 × 10^−4^
Metabolism of xenobiotics by cytochrome P450	−1.10	(−1.14–−1.07)	−1.16	(−1.19–−1.13)	−1.15	(−1.18–−1.13)	8.91 × 10^−3^
Bile secretion	−1.10	(−1.13–−1.07)	−1.14	(−1.16–−1.12)	−1.13	(−1.15–−1.11)	1.37 × 10^−2^
Fatty acid degradation	−1.09	(−1.13–−1.06)	−1.21	(−1.24–−1.18)	−1.19	(−1.21–−1.17)	1.50 × 10^−7^
Porphyrin and chlorophyll metabolism	−1.09	(−1.13–−1.05)	−1.17	(−1.20–−1.13)	−1.13	(−1.16–−1.11)	3.74 × 10^−2^
Butanoate metabolism	−1.06	(−1.10–−1.03)	−1.14	(−1.17–−1.11)	−1.12	(−1.15–−1.10)	7.69 × 10^−3^
Valine, leucine and isoleucine degradation	−1.05	(−1.09–−1.02)	−1.18	(−1.20–−1.15)	−1.14	(−1.16–−1.12)	4.24 × 10^−8^
Tryptophan metabolism	−1.05	(−1.08–−1.02)	−1.08	(−1.11–−1.06)	−1.09	(−1.11–−1.07)	8.57 × 10^−3^
Tyrosine metabolism	−1.05	(−1.09–−1.01)	−1.11	(−1.14–−1.08)	−1.09	(−1.12–−1.07)	0.139
Taste transduction	−1.05	(−1.07–−1.04)	−1.05	(−1.06–−1.04)	−1.05	(−1.05–−1.04)	0.453
Phototransduction	−1.04	(−1.06–−1.02)	−1.07	(−1.09–−1.06)	−1.06	(−1.08–−1.05)	3.09 × 10^−2^
Ovarian steroidogenesis	−1.04	(−1.06–−1.01)	−1.03	(−1.05–−1.01)	−1.04	(−1.05–−1.02)	0.967
Cardiac muscle contraction	−1.03	(−1.05–−1.02)	−1.06	(−1.07–−1.05)	−1.05	(−1.06–−1.04)	4.24 × 10^−9^
Pantothenate and CoA biosynthesis	−1.03	(−1.07–1.00)	−1.03	(−1.06–−1.00)	−1.03	(−1.05–−1.01)	1.18 × 10^−2^
beta-Alanine metabolism	1.01	(−1.02–1.04)	−1.03	(−1.05–−1.01)	−1.03	(−1.05–−1.01)	4.25 × 10^−2^
Folate biosynthesis	1.02	(−1.01–1.06)	−1.03	(−1.06–−1.00)	−1.01	(−1.03–1.01)	1.30 × 10^−3^
Primary bile acid biosynthesis	1.08	(1.03–1.14)	−1.01	(−1.05–1.02)	−1.01	(−1.04–1.02)	1.66 × 10^−2^
Riboflavin metabolism	1.09	(1.02–1.17)	−1.03	(−1.08–1.02)	1.01	(−1.03–1.05)	2.98 × 10^−3^

**Table 2 cancers-15-02921-t002:** Differential expression of the 28 colon-inflammation-related pathways (identified from US sporadic CRC patients), replicated in BD sporadic CRC patients stratified by MSI status.

Pathway (GO Description)	MSI: CRC vs. Normal		MSS: CRC vs. Normal		Interaction *p*
	FC	(95% CI)	FC	(95% CI)	
Nitrogen metabolism	−1.72	(−1.89–−1.56)	−1.70	(−1.80–−1.60)	0.232
Sulfur metabolism	−1.36	(−1.44–−1.28)	−1.26	(−1.31–−1.21)	2.03 × 10^−2^
Proximal tubule bicarbonate reclamation	−1.36	(−1.42–−1.30)	−1.33	(−1.37–−1.30)	0.069
Pentose and glucuronate interconversions	−1.35	(−1.40–−1.29)	−1.23	(−1.26–−1.20)	3.37 × 10^−6^
Synthesis and degradation of ketone bodies	−1.32	(−1.41–−1.24)	−1.19	(−1.24–−1.15)	4.83 × 10^−3^
Ascorbate and aldarate metabolism	−1.29	(−1.35–−1.23)	−1.20	(−1.23–−1.16)	8.54 × 10^−7^
Retinol metabolism	−1.27	(−1.31–−1.23)	−1.20	(−1.22–−1.18)	1.78 × 10^−4^
Steroid hormone biosynthesis	−1.22	(−1.25–−1.18)	−1.14	(−1.16–−1.12)	1.80 × 10^−4^
Drug metabolism-cytochrome P450	−1.22	(−1.25–−1.19)	−1.17	(−1.19–−1.15)	2.32 × 10^−4^
Fatty acid degradation	−1.21	(−1.23–−1.18)	−1.17	(−1.19–−1.15)	5.42 × 10^−11^
Chemical carcinogenesis	−1.18	(−1.21–−1.16)	−1.15	(−1.16–−1.13)	6.15 × 10^−5^
Porphyrin and chlorophyll metabolism	−1.17	(−1.21–−1.13)	−1.12	(−1.14–−1.10)	4.22 × 10^−5^
Metabolism of xenobiotics by cytochrome P450	−1.17	(−1.20–−1.14)	−1.14	(−1.15–−1.12)	1.78 × 10^−4^
Bile secretion	−1.16	(−1.18–−1.13)	−1.12	(−1.14–−1.11)	3.57 × 10^−2^
Valine, leucine and isoleucine degradation	−1.15	(−1.18–−1.12)	−1.13	(−1.14–−1.11)	1.96 × 10^−16^
Mineral absorption	−1.15	(−1.18–−1.12)	−1.17	(−1.19–−1.15)	0.193
Butanoate metabolism	−1.14	(−1.17–−1.11)	−1.11	(−1.13–−1.09)	1.17 × 10^−4^
Tyrosine metabolism	−1.11	(−1.14–−1.08)	−1.08	(−1.10–−1.06)	0.198
Phototransduction	−1.08	(−1.10–−1.06)	−1.06	(−1.07–−1.05)	0.100
Tryptophan metabolism	−1.07	(−1.10–−1.05)	−1.08	(−1.09–−1.06)	1.74 × 10^−5^
Cardiac muscle contraction	−1.05	(−1.06–−1.04)	−1.05	(−1.05–−1.04)	1.00 × 10^−5^
Folate biosynthesis	−1.04	(−1.07–−1.01)	1.00	(−1.02–1.02)	9.67 × 10^−3^
Riboflavin metabolism	−1.04	(−1.10–1.02)	1.03	(−1.01–1.06)	0.079
Taste transduction	−1.04	(−1.06–−1.03)	−1.05	(−1.06–−1.04)	0.366
Ovarian steroidogenesis	−1.04	(−1.06–−1.02)	−1.04	(−1.05–−1.02)	0.930

## Data Availability

All supporting data are presented in the tables presented in the main manuscript and as Appendix A.

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
