# Peer review of "Pathways Related to Colon Inflammation Are Associated with Colorectal Carcinoma: A Transcriptome- and Methylome-Wide Study"

_cancers, 2023, doi:10.3390/cancers15112921_

Round 1

Reviewer 1 Report

In this work, authors identified a number of dysregulated metabolic and other pathways in ulcerative colitis (UC), and demonstrated that many of those inflammation related gene pathways were also associated with sporadic colorectal carcinoma (CRC). The findings from US patients were also replicated in Bangladeshi patients with CRC. They demonstrate that this “inflammation – cancer” association was independent of sex and conventional genetic factors. Authors performed the RNA sequencing. Genome-wide gene expression assay, Genome-wide methylation assay, Microsatellite instability (MSI) detection, KRAS mutation detection, and huge statistical analyses.

 Subject definitions and methods are optimal. Figures and references are adequate.

I have found this paper relevant to the field of this journal. I have only but many minor comments.

Minor points:

1. Introduction:

1) The line 113, put the full name of CIMP and then this acronym in parenthesis.

2) The line 120, correct a word “prometer” into “promoter”.

3) The lines 121-124, check the sentence, its meaning is not clear.

2. Materials and Methods:

4) Subtitles, keep the same style: Only the first letter is capital: Genome-wide gene expression assay (line 179), Microsatellite instability (MSI) detection (line 201), and Statistical analysis (line 207).

3. Results:

5) The line 291, put the full name of KEGG and then this acronym in parenthesis.

6) The line 312, put words in the following order “… the expression profile of cancerous colonic mucosa (Sporadic CRC) from the sporadic CRC patients …”

7) In the subchapters 3.1., 3.2., 3.3., change all abbreviation “Fig” into “Figure” (keep the same expression a word “Figure” for the whole article).

8) Similarly, keep the same expression “Supplementary Table/Figure” (with the capital letters S, T/F) for the whole article, correct this in the lines: 291, 307, 308, 315, 351, 368, 385, 386, 409, 413, 419 and 481.

9) Figures 2 (line 300), 4 (lines 344-345), 5 (line 360) and 10 (line 492), put “the y-axis” (include the definite article).

10) Figure 3 (line 325), remove the first “and” and leave the second one “… (b) UC CRC vs. Control (blue), (c) UC vs. Control (pink) and (d) UC CRC

11) In the description of Table 1 (line 390) and Table 2 (line 394), include “sporadic CRC”: … (identified from US sporadic CRC patients), replicated in BD sporadic CRC patients …

12) The line 400, keep the same style: stage 1, stage 2 and stage 3 (like in the lines 405-406 – remove the dash).

13) The line 405, correct number “20%” into “21%” (according Table 1).

14) The line 406, put in one parenthesis “(interaction p=1.5x10-7, see Table1)” (like in the lines 401, 408, 411 and 413) with a corrected number “p=1.5x10-7 (according Table 1).

15) The line 414, correct a word “these” (plural) into “this” (singular).

16) You have very nice Figure 7, but you do not mention it in the text!!!

17) In the subchapter 3.6. “Nitrogen metabolism, genes for Urea Cycle enzymes”, please check all names of substances and enzymes. Use the small letters for pyrimidine (line 432) and ornithine (line 434), correct words “citrulin” into “citrulline” (lines 434 and 440) and “argenine” into “arginine” (line 440).

18) In the Figure 8, put “the horizontal axis” (include the definite article in the line 445). Correct words “CRS1” into “CPS1” (line 445), “citrulin” into “citrulline” (lines 446 and 448) and “arginisuccinic acid” into “argininosuccinic acid” (lines 446-447). Use the small letters for arginine, citrulline, arginase1 and arginase (line 448). Put words “ASL”, “NOS3” (line 447) and “ARG” (line 449) in italics (keep the same writing of enzyme symbols for the whole subchapter 3.6. and the description of Figure 8).

19) The title of subchapter 3.7, correct a word “inflammaotry” into “inflammatory” (line 450).

20) In the subchapter 3.7, correct “Figure 8” into “Figure 9” (line 453), and put the comma after CRC “… in sporadic CRC, and Th1 and Th2 cell differentiation …” (line 459). Cancel writing in italics of “IL-17”, because it is an interleukin, not enzyme (lines 458 and 461).

21) In the Figure 9, correct a word “cytokine” into “chemokine” (line 464), and cancel writing in italics of a symbol “IL-17” (line 466).

22) In the subchapter 3.8, correct the second “reflects” into “reflect” (line 483) and a word “glucoronate” into “glucuronate” (line 486).

4. Discussion:

23) The line 508, cancel writing in italics of a word “and”.

24) The line 510, remove a word “and” in front of “tryptophan metabolism” and the comma after parenthesis.

25) The line 518, put the comma after a word “humans“.

26) The line 526, remove the comma after a word “phosphorylation“.

27) The line 527, cancel writing in italics of a word “ETHE1” (the same as in the line 524).

28) The line 530, cancel writing in italics of a word “TST” (the same as in the line 529).

29) The lines 534-563, cancel writing in italics of all words (CA.., AQP8, SLC4A4, NAT...).

Ad 27), 28), 29) – Their meaning here is more like a protein than like a gene.

30) The lines 579, 581, 582, change “figure 4…” into “Figure 4…” (with the capital letter).

I recommend this paper for acceptation after minor revision in the journal.

Moderate editing of English language.

Author Response

Response to Reviewer’s comment:

Comments and Suggestions for Authors:

In this work, authors identified a number of dysregulated metabolic and other pathways in ulcerative colitis (UC), and demonstrated that many of those inflammation related gene pathways were also associated with sporadic colorectal carcinoma (CRC). The findings from US patients were also replicated in Bangladeshi patients with CRC. They demonstrate that this “inflammation – cancer” association was independent of sex and conventional genetic factors. Authors performed the RNA sequencing. Genome-wide gene expression assay, Genome-wide methylation assay, Microsatellite instability (MSI) detection, KRAS mutation detection, and huge statistical analyses.

 Subject definitions and methods are optimal. Figures and references are adequate.

I have found this paper relevant to the field of this journal. I have only but many minor comments.

Response: We are extremely thankful to you for the kind comments and all the suggestions. We understand how minutely you have reviewed the manuscript and have suggested all the corrections. We are pleased to make all these corrections. Please note that in this revised version the line numbers have changed slightly. However, you will find that we have addressed all the points you have mentioned.

Minor points:

  1. Introduction:

1) The line 113, put the full name of CIMP and then this acronym in parenthesis. Response: Corrected.

2) The line 120, correct a word “prometer” into “promoter”. Response: Corrected.

3) The lines 121-124, check the sentence, its meaning is not clear. Response: we have now revised the sentence accordingly.

  1. Materials and Methods:

4) Subtitles, keep the same style: Only the first letter is capital: Genome-wide gene expression assay (line 179), Microsatellite instability (MSI) detection (line 201), and Statistical analysis (line 207).

Response: Corrected

  1. Results:

5) The line 291, put the full name of KEGG and then this acronym in parenthesis. Response: Thanks again. We corrected.

6) The line 312, put words in the following order “… the expression profile of cancerous colonic mucosa (Sporadic CRC) from the sporadic CRC patients …” Response: Thanks again. Corrected.

7) In the subchapters 3.1., 3.2., 3.3., change all abbreviation “Fig” into “Figure” (keep the same expression a word “Figure” for the whole article). Response: Corrected.

8) Similarly, keep the same expression “Supplementary Table/Figure” (with the capital letters S, T/F) for the whole article, correct this in the lines: 291, 307, 308, 315, 351, 368, 385, 386, 409, 413, 419 and 481. Response: Corrected.

9) Figures 2 (line 300), 4 (lines 344-345), 5 (line 360) and 10 (line 492), put “the y-axis” (include the definite article). Response: Corrected.

10) Figure 3 (line 325), remove the first “and” and leave the second one “… (b) UC CRC vs. Control (blue), (c) UC vs. Control (pink) and (d) UC CRC.

Response: Corrected.

11) In the description of Table 1 (line 390) and Table 2 (line 394), include “sporadic CRC”: … (identified from US sporadic CRC patients), replicated in BD sporadic CRC patients …

Response: Corrected.

12) The line 400, keep the same style: stage 1, stage 2 and stage 3 (like in the lines 405-406 – remove the dash). Response: Thanks again. Corrected

13) The line 405, correct number “20%” into “21%” (according Table 1). Corrected.

14) The line 406, put in one parenthesis “(interaction p=1.5x10-7, see Table1)” (like in the lines 401, 408, 411 and 413) with a corrected number “p=1.5x10-7” (according Table 1). Corrected

15) The line 414, correct a word “these” (plural) into “this” (singular). Corrected.

16) You have very nice Figure 7, but you do not mention it in the text!!!

Response: Thank you very much for noticing that. We have now mentioned this in the text.

17) In the subchapter 3.6. “Nitrogen metabolism, genes for Urea Cycle enzymes”, please check all names of substances and enzymes. Use the small letters for pyrimidine (line 432) and ornithine (line 434), correct words “citrulin” into “citrulline” (lines 434 and 440) and “argenine” into “arginine” (line 440). Response: Sorry for the mistakes. We now corrected.

18) In the Figure 8, put “the horizontal axis” (include the definite article in the line 445). Correct words “CRS1” into “CPS1” (line 445), “citrulin” into “citrulline” (lines 446 and 448) and “arginisuccinic acid” into “argininosuccinic acid” (lines 446-447). Use the small letters for arginine, citrulline, arginase1 and arginase (line 448). Put words “ASL”, “NOS3” (line 447) and “ARG” (line 449) in italics (keep the same writing of enzyme symbols for the whole subchapter 3.6. and the description of Figure 8). Response: Corrected.

19) The title of subchapter 3.7, correct a word “inflammaotry” into “inflammatory” (line 450). Response: Thanks for pointing this, We have corrected this.

20) In the subchapter 3.7, correct “Figure 8” into “Figure 9” (line 453), and put the comma after CRC “… in sporadic CRC, and Th1 and Th2 cell differentiation …” (line 459). Cancel writing in italics of “IL-17”, because it is an interleukin, not enzyme (lines 458 and 461). Response: Corrected.

21) In the Figure 9, correct a word “cytokine” into “chemokine” (line 464), and cancel writing in italics of a symbol “IL-17” (line 466). Corrected.

22) In the subchapter 3.8, correct the second “reflects” into “reflect” (line 483) and a word “glucoronate” into “glucuronate” (line 486). Response: Corrected.

  1. Discussion:

23) The line 508, cancel writing in italics of a word “and”. Response: Corrected.

24) The line 510, remove a word “and” in front of “tryptophan metabolism” and the comma after parenthesis. Response: Corrected.

25) The line 518, put the comma after a word “humans“. Response: Corrected.

26) The line 526, remove the comma after a word “phosphorylation“. Response: Corrected.

27) The line 527, cancel writing in italics of a word “ETHE1” (the same as in the line 524). Response: Corrected.

28) The line 530, cancel writing in italics of a word “TST” (the same as in the line 529). Response: Corrected.

29) The lines 534-563, cancel writing in italics of all words (CA.., AQP8, SLC4A4, NAT...). Response: Corrected.

Ad 27), 28), 29) – Their meaning here is more like a protein than like a gene. ***

30) The lines 579, 581, 582, change “figure 4…” into “Figure 4…” (with the capital letter). Response: Corrected.

I recommend this paper for acceptation after minor revision in the journal.

Response: Thank you very much for the recommendation. We really appreciate all your help.

Reviewer 2 Report

The manuscript is very well written and prepared with scientifically interesting findings. I have only some minor comments:

1. Figure 2 and 4 are missing the section labels (A,B,C,...). They are presented in the text, but not in the figure nor figure caption. Also, there is a typo "genen->genes" in the title of Fig.2B. I suggest to change x-axis labels (gene names) as vertical text, so there would be more gene IDs visible.

2. Table title S1 and S7 have typos --> "patient".

3. Is "deep sea" (p.4 l.191) same than "open sea"? In my opinion, "open sea" is more commonly used one.

4. Table S1:

a. Under "Tumor grade", there is "low" and "high" - could you clarify how to made up this two-scale grading system?

b. Also, it looks like under the same title, there is histological types of CRC - please, modify under the correct title and check numbers. Now there is 6 mucinous tumors, but what about 4 others?

c. Under "Tumor multiplicity" there is 3+4 tumors, what about rest 3? Can you clarify, what do you mean with "Tumor multiplicity" in here?

d. Under "Ethnicity" there is only "hispanic 1"- rest 9? Can you try to combine Race and Ethnicity? In my opinion, it would be slightly better to present everything under the title  of "Ethnicity".

e. It would be good to know also patients' age range besides mean (of age).

f. In general, could you make Table S7 and S1 similar and present the same characteristics under the same titles?

5. It would be interesting to also see the genelists associated the pathways that were found to be altered (i.e. those differentially expressed genes that made those pathways altered). Could you provide it e.g. as a supplementary table?

Author Response

Response to Reviewer:
We are extremely thankful for the kind comments and all the suggestions. We understand how minutely you have reviewed the manuscript and have made the very helpful suggestions. We have considered all your suggestions while revising this manuscript. We hope that you would find improvement in this revised version. 
Please note that in this revised version the line numbers have changed slightly. However, you will find that we have addressed all the issues point by point.
Comment:
The manuscript is very well written and prepared with scientifically interesting findings. I have only some minor comments:
Response: Thank you very much for the kind comment. We appreciate that you liked this work.

1.    Figure 2 and 4 are missing the section labels (A,B,C,...). They are presented in the text, but not in the figure nor figure caption. Also, there is a typo "genen->genes" in the title of Fig.2B. I suggest to change x-axis labels (gene names) as vertical text, so there would be more gene IDs visible. *** Also, revise title of Figure 4A to start with “Differential expression…” instead of “Differential expressed…” 

Response: Thanks for noticing this. Now we have included that and corrected the typo. We took your suggestion and made changes in the x-axis gene name orientation. While trying to put the gene names vertically, we noticed that  some of the gene names were cut. So we used 45 degree orientation for Figure 2A and 2B. There were too manty genes in Figure 2C, D and E. There was no way to accommodate all the gene names. However as per your suggest#5, we have added a Supplemental Table (S13) where the list of significant genes that contribute to the overall dysregulation of a given pathway, is shown. Hope that would help.

2. Table title S1 and S7 have typos --> "patient". 

Response: Thanks for noticing. We are pleased to correct that.

3. Is "deep sea" (p.4 l.191) same than "open sea"? In my opinion, "open sea" is more commonly used one. 

Response: Yes, your assumption is correct. “deep sea” and “open sea” mean the same. We have no problem accepting your suggestion. We have now changed to “open sea”.

4. Table S1:
a. Under "Tumor grade", there is "low" and "high" - could you clarify how to made up this two-scale grading system? 
Response: Well-differentiated tumors were grouped as “low” grade and moderate to poorly differentiated tumors were grouped as “high” grade. We have mention this in the method section now.
b. Also, it looks like under the same title, there is histological types of CRC - please, modify under the correct title and check numbers. Now there is 6 mucinous tumors, but what about 4 others? 
Response: Thanks for noticing that. We have corrected this in the revised Table S1.
c. Under "Tumor multiplicity" there is 3+4 tumors, what about rest 3? Can you clarify, what do you mean with "Tumor multiplicity" in here? Response: We have deleted that.
d. Under "Ethnicity" there is only "hispanic 1"- rest 9? Can you try to combine Race and Ethnicity? In my opinion, it would be slightly better to present everything under the title  of "Ethnicity". 
Response: Your suggestion is well taken and we have now presented only one variable.
e. It would be good to know also patients' age range besides mean (of age). Response: Thanks for mentioning that. We agree and therefore in this revised version, we have included the SD along with the mean.
f. In general, could you make Table S7 and S1 similar and present the same characteristics under the same titles? 
Response: We have now tried to put the information in similar format. However, some of the information was not available for the US patients and therefore we could not make exactly the same format for Table S1 and S7. Sorry for that.
5. It would be interesting to also see the gene lists associated the pathways that were found to be altered (i.e. those differentially expressed genes that made those pathways altered). Could you provide it e.g. as a supplementary table? 
Response: Thanks for the suggestion. We have now added another Supplemental Table and have included that information in new Supplemental Table S13 in this revised version.